# The Role of Self-Compassion and Attributions in the Mental Health of Older Adolescents amid the COVID-19 Pandemic

**DOI:** 10.3390/ijerph20216981

**Published:** 2023-10-27

**Authors:** Jelena Maricic, Sila Bjelic, Katarina Jelic

**Affiliations:** Faculty of Croatian Studies, University of Zagreb, 10000 Zagreb, Croatiakjelic@hrstud.hr (K.J.)

**Keywords:** self-compassion, attributional styles, mental health, older adolescents, COVID-19 pandemic

## Abstract

This study aimed to examine the relationship among self-compassion, attributional styles, and mental health and their components in older adolescents in the context of the COVID-19 pandemic. The role of each component of self-compassion (self-kindness, common humanity, mindfulness, self-judgment, isolation, and over-identification) and attributions (globality, stability, self-worth, and negative consequences) in predicting mental health was also analyzed. There were 322 participants aged 18 to 22 that participated in an online survey. The participants filled out a form that consisted of sociodemographic questions, COVID-19-related questions, the Self-Compassion Scale, the Mental Health Continuum—short form—and the Cognitive Styles Questionnaire—very short form. The results indicated moderate levels of self-compassion, attributions, and mental health in participants. Furthermore, gender differences in self-compassion were confirmed, meaning that male participants had higher total levels of self-compassion, and certain differences were observed on attribution subscales, but not on well-being subscales. Self-compassion and mental health were found to be positively correlated with each other and negatively correlated with negative attributions. Of the four attributional components, stability and negative consequences were revealed to be significant negative predictors in the first step but lost their significance with the inclusion of self-compassion components in the second step of the analysis. Regarding the six components of self-compassion, self-kindness, recoded isolation, and common humanity were significant positive predictors in the second step of the analysis. COVID-19-related items did not show any significant intergroup differences. Our findings contribute to a better understanding of the relationship between positive mental health, self-compassion, and attributions in older adolescents so that they can be used as theoretical support for related interventions, especially during and after times of crisis, such as a pandemic.

## 1. Introduction

With the increasing uncertainty of pandemic conditions and forced shifts in how adolescents communicate, learn, and develop, findings regarding the importance of self-compassion and attribution styles in predicting mental health problems in the youth must be reassessed and further explored. The coronavirus (COVID-19) pandemic started with the outbreak in China in 2019, and as of May 2023, it has affected roughly 687 million people globally, causing almost 6.87 million deaths [1]. Besides health and economic crises, scientists also pointed out the pandemic’s adverse psychological effects [2], and adolescents were identified as one of the most vulnerable groups [3]. The central premise of this study refers to the protective role of self-compassion and its potential to decrease the stress of older adolescents during the COVID-19 pandemic [4] and promote positive thinking, all of which diminish mental health problems.

### 1.1. Mental Health

Keyes [5] defines mental health as a combination of high levels of emotional, psychological, and social well-being. Additionally, the World Health Organization [6] states that to retain mental health, people must be productive and contribute to their community, which was much more difficult to achieve during the pandemic, combined with difficulties in experiencing various pleasures that include other people [7].

Mental health and mental illness have often been portrayed as opposites, but they are certainly not mutually exclusive. Each of these constructs has its own continuum, so it is possible for an individual to have low mental health without any symptoms of illness and vice versa [8,9]. Shin and Lim [7] differentiate three dimensions of positive mental health: emotional (e.g., happiness), psychological (e.g., self-realization), and social (social value) well-being, which are influenced by situational determinants, both in healthy and in mentally ill populations [10].

Croatian students previously self-reported low to medium levels of mental health, with only 3.7% achieving results that could be classified as a high level of mental health [11]. Several studies have claimed to detect poorer mental health in girls, especially those that do not have adequate social support [12]. Reports during the pandemic have estimated that girls experience more psychological distress, less life satisfaction, and higher results on syndrome and internalizing broadband scales [13,14]. Vučenović et al. [15] found that amid the COVID-19 crisis, girls have reported higher levels of depression, while others state their symptoms are more severe [16]. Some authors [17] advocate the notion that such differences are maintained throughout life, except for suicide, which is more common in men without sufficient social support [12]. Adult men can also be more vulnerable in situations of economic crisis, especially during young adulthood [18].

### 1.2. Self-Compassion

Close to the concept of self-compassion is the concept of self-esteem, with several key differences among them. Self-esteem has long been established as a key concept in psychological well-being due to its relations with both positive and negative psychological outcomes, such as depression and anxiety [19], social isolation [20], substance use [21], disordered eating behaviors [22], and other health-related behaviors [23,24].

Unlike self-compassion, self-esteem is founded on comparing oneself to others or to some normative standard, while self-compassion is not based on such judgmental evaluations. Additionally, self-esteem is primarily based on cognitive assessment, which can contribute to activities to increase and maintain it, but does not necessarily include coping strategies, while self-compassion actively aims to reduce one’s own suffering via, for example, cognitive reframing. Moreover, self-esteem is more related to the sympathetic (threat reaction) nervous system, while self-compassion is more related to the parasympathetic (soothing) nervous system [25]. Pandey et al. confirmed positive correlations between self-esteem, self-compassion, and well-being [26].

In an attempt to preserve, maintain, and improve their self-esteem, individuals can be subjected to numerous problematic decisions and behaviors that could lead to narcissism and violence [27]. Many psychological theories proclaim how the pursuit of maintaining positive emotions, including high self-esteem, can lead to cognitive biases and distortions, self-deception, or favoritism. In contrast, self-compassion offers little incentive for false beliefs and defensiveness, since accepting our weak points is portrayed as a success rather than failure [28]. Hence, self-compassion could potentially co-op the position of self-esteem in the theoretical and empirical discourse regarding mental health.

Neff operationalized self-compassion as a construct with three main components: self-kindness versus self-judgment (meaning accepting reality with compassion and gentleness), common humanity (seeing one’s personal experiences as part of a broader human experience versus isolation), and mindfulness (focused awareness of personal suffering versus identifying with it) [29]. Self-compassion must not be mistaken for self-pity (when relating to painful experiences) or self-indulgent or selfish behavior. It is found that women tend to have significantly less self-compassion compared to men [30]. Murn and Steele found that chronological age did not mitigate gender differences in young adults on the overall self-compassion level, but age does appear to be a factor in an individual’s ability to maintain a mindful, shared perspective when facing emotional or cognitive distress [31]. Adolescents’ egocentrism and “personal fables” incline them to believe that their experiences are unique and failures are fatal rather than a normal part of life, which is a belief that “fades” during maturation, but during adolescence, it can affect behavior [32].

Self-compassion is linked to self-care practices, such as seeking professional help when in need, proactivity, or having a healthy lifestyle [33]. The literature suggests that the absence of self-care can lead to higher perceived stress levels [34], and those with a lower mental health status are found to be less likely to engage in self-care activities [35]. This absence of self-care mediated the relationship between the perceived COVID-19 threat and the fear of dying during the pandemic [36]. Furthermore, individuals with higher self-compassion had a higher tolerance for uncertainty, effectively decreasing the fear of COVID-19. It is also shown that self-compassion moderates the negative effect of COVID-19 information overload on the fear of COVID-19 [37].

### 1.3. Attributions

The experience of personal suffering and failure is strongly associated with the attribution process. Attributions help us explain behaviors and their outcomes. Weiner’s original model of attributions [38] underlies much of the research on the cognitive, behavioral, and emotional consequences of attributional processes. It presumes three dimensions:Internality vs. externality, i.e., attributing a behavior to either a person or an environment;Stability vs. instability, i.e., whether the perceived cause of the behavior is constant or variable;Globality vs. specificity, i.e., whether the perceived cause is specific to a particular behavior or appears in general in various situations in a person’s life.

People with a more negative attribution style tend to ascribe unpleasant events to internal, stable, and global causes but positive events to external, unstable, and specific causes, which, after numerous aversive events that we perceive to be uncontrollable, can lead to the deterioration of physical and mental health and eventually to a state of learned helplessness [39]. The outcomes of learned helplessness vary from a lack of commitment to severe depression, particularly regarding self-generated stressors [40].

The theory of helplessness is upgraded into the theory of hopelessness by emphasizing the importance of negative expectations and negative self-worth [41]. Helplessness provides an internal, permanent, and global attribution of adverse events, implying a person’s inability or inadequacy, leading to behavioral disengagement. The harmful effects of negative attributions are aligned with an individual’s well-being [42]. Self-compassion promotes healthier attributional styles [43], mitigates unpleasant emotions, and protects an individual’s well-being [44]. Among college students, negative life events were found to be related to greater hopelessness and to more suicidal behavior, but self-compassion was seen to attenuate this effect [45]. This result can be seen in the light of findings that more self-compassionate people are more inclined to feel appropriately responsible for their thoughts and actions, meaning that they are not prone to self-blame, but to positive reframing and self-forgiveness [46] since kindness toward oneself is one of the most important predictors of positive mental health [7]. Regarding social well-being, college students with a pessimistic attributional style feel unable to affect their social environment positively and refuse to make an effort to approach others [47].

The COVID-19 pandemic changed our perceived control over our lives immensely. In a study by Skarżyńska et al., the respondents tended to attribute more responsibility for COVID-19 effects to the government than to other (“non-government”) factors, which was a new and frightening form of causal attribution [48]. At the same time, Kotera et al. stated that self-compassion moderates motivation, in a way that means that higher self-compassion can help transfer motivation from extrinsic to intrinsic, which was a very important issue during a period of significant reduction in external stimuli [49].

### 1.4. Aim of This Research

Worldwide studies have shown self-compassion’s significant and practical value, suggesting benefits from interventions that raise self-compassion [29] or from including self-compassion in psychotherapy [50] and the treatment of depression, anxiety, and personality disorders. The relationship between self-compassion, mental health, and negative attributions in the Croatian adolescent population has not yet been investigated.

As such, this paper aims to determine the contribution of attributional styles and components of self-compassion in understanding adolescents’ mental health during the COVID-19 crisis, a period of great uncertainty linked to many psychological conditions and emotional responses such as fear, panic, and depression [51,52]. More specific goals include determining the levels of positive mental health (well-being), attribution dimensions, and self-compassion in older adolescents, while exploring gender- and COVID-19-status-related differences in the later phase of the pandemic.

## 2. Materials and Methods

### 2.1. Participants

A total of 333 participants took part in this study, though 11 uncompleted forms were excluded from further analysis. Of the 322 participants, 276 (76.4%) identified as female and 46 (23.6%) identified as male. The age range was from 18 to 22 years, and the average age of the participants was 19 years (M = 19.03, SD = 1.35). Most of the participants, 61.5%, were high school students, followed by non-working (29.5%) and working (6.8%) university students, followed by employed (1.2%) and unemployed (0.9%) non-student youths. The participants were mainly residents of the capital Zagreb (52.2%), followed by cities with less than 500,000 residents (37.6%) and cities with less than 2000 inhabitants (10.2%), resulting in a total of 11 cities.

### 2.2. Questionnaires

Our questionnaire consisted of sociodemographic questions, namely, gender, age, employment status, and the size of the place of residence. Additionally, we included the Self-Compassion Questionnaire [53], the Mental Health Continuum—short form [54]—and the Cognitive Styles Questionnaire—very short form [55]. Apart from that, we included six COVID-19-related questions, to which the participants answered “yes” or “no” (a dichotomous scale):Have you or has someone close to you had the coronavirus?If so, did you have any health consequences as a result?Are you or is someone close to you in a high-risk group for contracting the coronavirus?Have you or has someone close to you been hospitalized due to the coronavirus?Has someone close to you died from the coronavirus?Have you or has someone close to you lost their job due to the coronavirus?

This particular scale had a low reliability score of 0.57, which is still considered acceptable by several authors [56], provided that results are interpreted with caution.

#### 2.2.1. The Self-Compassion Scale

The Self-Compassion Scale [53] consists of 26 items that express the typical behavior of individuals toward themselves under challenging circumstances, divided into six subscales that follow the authors’ theoretical framework [53]: gentleness toward oneself (5 items), self-judgment (5 items), and shared humanity, isolation, focused awareness, and overidentification (each with 4 items). The author examined the scale in 20 diverse samples, and in each of them, a general self-compassion factor and six specific factors were supported [57]. Participants estimate how much each item applies to them by choosing a number from 1 (almost never) to 5 (almost always). An example of a statement related to shared humanity reads: When I feel depressed, I remember that there are many people who feel the same way as me. The result on each subscale is the mean value of the pertaining items, where higher scores indicate higher levels of self-compassion. The total score reliability coefficient in this study was 0.88, ranging from 0.72 to 0.83 between the subscales.

#### 2.2.2. The Cognitive Styles Questionnaire

The Cognitive Styles Questionnaire—very short form [55]—was developed by modifying and expanding the Attributional Styles Questionnaire, which measures the internality, globality, and stability of an individual’s attributions using six positive and six adverse hypothetical events. Later iterations included self-worth implications and negative consequences subscales [58]. A very short form [55] was created during the German validation with sufficient psychometric properties of 0.91 with 27 items, and six negative scenarios measuring the globality (7 items) and stability (7 items) of attributions, as well as self-worth implications (10 items) and negative consequences (3 items). The participant’s task is to imagine described events, think of possible causes, and answer how much they agree with suggested statements from 1 (I completely disagree) to 5 (I completely agree). Results can range from 27 to 135, and a higher score on the questionnaire indicates a more negative cognitive attribution style. The Croatian translation was obtained by three people (a native English speaker, a psychologist, and a layman) translating the CSQ-SF from English. The total score reliability coefficient in this research was 0.87, ranging from 0.53 to 0.81 between the subscales. The scale has many purposes, such as measuring changes in cognitive style as a result of specific treatments for depression [59].

#### 2.2.3. Mental Health Continuum Scale

The Mental Health Continuum—short form [54]—consists of 14 items that examine positive mental health on a Likert scale where 0 means “never” and 5 means “every day”. The longer version consists of 40 items with three factors (emotional, psychological, and social well-being), whereas the short form offers only 14 with high reliability (0.94 in this research; ranging from 0.86 to 0.90 between the subscales); 3 of them measure emotional well-being, 6 psychological well-being, and 5 social well-being. Combined, they represent the concept of eudemonic well-being, or subjective experiences. The total result for each subscale is the mean value, and higher scores indicate greater levels of positive well-being. Each item starts the same (In the past three months, how often did you feel); for example, an emotional well-being item ends in satisfaction with life. The participants must then mark the frequency of these feelings. Confirmatory analysis confirmed the existence of 3 subscales, corresponding to the assumed factor structure from the original model. The MHC Scale has been used in many COVID-19-related studies [60,61].

### 2.3. Procedure

After obtaining the Institutional Review Board’s approval, research was conducted online using social networks such as Facebook (the most used social network in Croatia at the time [62]), students’ mailing lists (e.g., of student dormitories), and the snowball technique (which included the help of high school psychologists or older high school students active in different groups). The only inclusion/exclusion criterion was age (18–22). All potential participants were informed about the purpose of this study. Their involvement was anonymous and voluntary, without any incentive or reward, and they could withdraw from the study at any given time, about which they were also informed, so their answering the questionnaire was considered consent. Detailed instructions were provided for each scale, and researchers left their contact list for further questions or comments.

### 2.4. Data Analysis

We used the IBM^®®^ SPSS^®®^ Statistics software, version 29 (IBM, New York, NY, USA), which was released in September of 2022. The analysis included descriptive statistics and frequencies, Cronbach’s alpha reliability coefficient, the Kolmogorov–Smirnov goodness-of-fit test, and Levene’s variance test prior to performing the *t*-test for independent samples; after that, we conducted correlation (Pearson) and hierarchical linear regression analysis.

## 3. Results

### 3.1. Descriptive Data

We conducted a descriptive analysis for self-compassion, attributions, and mental health variables and inspected the preconditions for parametric procedures. Since the asymmetry values of the distribution in this sample did not exceed 0.6, and the flattening values did not exceed 0.9 (see Table 1), the distributions can be considered normal for parametric procedures. The results are presented in Table 1.

The mean of the total score on the Self-Compassion Scale was near the average theoretical values. The author suggests that an average overall self-compassion score is around 3.0; 2.5–3.5 is moderate; 1–2.5 is low; and above 3.5 is high. The highest average score on the Cognitive Styles Questionnaire was on the stability subscale, and the average results on the entire scale indicate moderately negative attributions. The emotional and psychological well-being of the participants show average values, but the social well-being is slightly above average. According to the scoring manual, 53.4% of our participants were within the normal range, 32.9% were flourishing, and 13.7% were not.

### 3.2. Gender Differences in Self-Compassion, Cognitive Styles, and Well-Being

Leven’s homogeneity of variance test revealed a statistically significant difference only in mental health variances (F = 7.278, *p* < 0.01). The next step was to perform a *t*-test for independent samples and analyze gender differences in the studied variables. The results are shown in Table 2.

A statistically significant difference in global measures of self-compassion is evident between boys and girls; girls showed a lower level of self-compassion. Analyzing individuals’ subscales on the Self-Compassion Scale, we found that boys achieved statistically significantly higher levels of mindfulness (T = −2.242, *p* < 0.05) and lower levels of over-identification (T = −4.713, *p* < 0.01) compared to girls. On the Cognitive Styles Questionnaire, further analysis showed that boys leaned toward more global attributions (T = −2.401, *p* < 0.05), expected more negative consequences (T = −2.357, *p* < 0.05), were inclined to perceive negative events as stable (T = −2.035, *p* < 0.01), and had overall higher scores on the questionnaire, indicating a more negative cognitive attribution style. Finally, the analysis of mental health components showed no statistically significant gender differences in emotional, social, or psychological well-being.

### 3.3. COVID-19 Status, Self-Compassion, Cognitive Styles, and Well-Being

The majority of our sample had experienced COVID-19, and 15.7% of those participants had suffered significant health consequences. Most participants (or their family members) were considered to be in the high-risk group for contracting COVID-19, some had experienced the hospitalization of (a) family member(s), and only a minority had lost someone close to them due to the COVID-19 disease. Additionally, almost a fifth of our sample answered ‘‘yes’’ when asked whether they or an immediate family member had lost a job due to COVID-19 restrictions (Table 3).

The COVID-19-related variables provided very few findings in terms of self-compassion, attributions, and mental health levels. A Mann–Whitney U nonparametric test for independent samples was used due to sample size inequalities [63], asserting that the participants who experienced job loss due to the pandemic had a lower level of self-compassion (Table 4). This point is elaborated on in the discussion section.

### 3.4. The Relationship between Self-Compassion, Cognitive Styles, and Well-Being

Prior to performing a regression analysis, we sought to inspect a correlation matrix for our predictor variable subscales (self-compassion and attribution) and the criterion variable, i.e., well-being. The results are presented in Table 5 and Table 6.

The results of the bivariate correlation analysis (Table 5) show a significant positive association between self-compassion and mental health, but a negative association with negative attributions. The correlation between mental health and negative attributions was significantly negative; participants with higher scores on the Cognitive Styles Questionnaire showed less satisfactory mental health.

As seen in Table 6, mental health is positively correlated with all components of self-compassion and negatively correlated with all dimensions of attribution. Mental health was positively correlated with dimensions of self-compassion and negatively with cognitive styles. In a practical sense, that would indicate that participants with higher scores on the subscales of globality, negative consequences, self-worth implications, and stability had a lower level of mental health. The most intercorrelated predictors are globality and negative consequences, kindness toward oneself, mindfulness, globality, and self-evaluation. With adequate correlations, we proceeded with our final specific aim and prediction analysis, as shown in Table 7 and Table 8.

The regression analysis model included predictor variables (four attribution subscales in the first step of the analysis and six components of self-compassion in the second step). A model including only attributions explained 20% of the total mental health variance, with stability and negative consequences as the only significant factors. In the second step, self-compassion dimensions were added, with an additional 20.2% ∆R, based on self-kindness, isolation, and common humanity. The combined model predicted 40.2% of the total variance in mental health. Significant predictors from the first step were not relevant in the second step, suggesting that attributions might not be directly associated with mental health status.

## 4. Discussion

This paper aimed to determine the contribution of attributional styles and components of self-compassion in explaining mental health among older adolescents, emphasizing any potential gender differences in the chosen constructs in a sample of Croatian older adolescents during the later phase of the COVID-19 pandemic.

### 4.1. Levels of Self-Compassion, Well-Being, and Attributions in the COVID-19 Pandemic Context

Moderate scores were achieved in most self-compassion dimensions. Mental health was slightly above the theoretical mean. The average results on the entire Cognitive Styles Questionnaire imply a tendency for relatively positive attributions in our sample.

Our participants scored lower on the social well-being component than on the emotional and psychological components, which may indicate that they felt, to a degree, isolated. We may find possible explanations for these observed differences in Neff’s model [29] of self-compassion, including three main dimensions: self-kindness vs. self-judgment, common humanity vs. isolation, and mindfulness vs. over-identification. Self-compassion leads to understanding our failures without subduing to self-criticism and, finally, to acceptance or even gratitude. Fighting against the circumstances only promotes stress and suffering, which could entice a feeling of isolation, as if we are the only one’s suffering. Self-compassion implies understanding the shared human experience and the scope of it. It also presumes that we steadily approach our emotions, seek a larger perspective, and be open to new experiences, non-judgmental, and present in the moment. The opposite of that is over-identifying with our thoughts and emotions as if they are the entire reality. The coronavirus pandemic represented a specific context for mental health research, but also to observe all the above-mentioned dimensions—how people tend to respond to insecurities and restrictions [64]. Being unable to leave the house made some individuals feel anxious and lonely, while others revisited older hobbies and crafts.

Buljubašić and Bulut [65] stated that people with low self-compassion do not give themselves the empathy they would give to a friend in the same situation, which may mean that those who believe that it is their “duty” to have a control over their lives are stricter with themselves. The pandemic is a situation in which everyone must accept that their control over their own life is reduced. On the other hand, perceived control plays an important role in people’s experiences. For example, the perception that “chance” was the controlling factor predicted longer unemployment in cases of unemployment and reemployment [66]. Therefore, the “correct” attribution of responsibility in difficult social circumstances such as a pandemic becomes more important than in regular circumstances. The acceptance of difficulties as part of a shared experience, especially during ongoing shared trauma such as that of the COVID-19 pandemic, is proven to be an empowerment factor [67].

### 4.2. Gender Differences in Studied Variables

Gender differences in observed mental health were not found in our research, in contrast to the predominant finding of previous research that young women are more vulnerable in the situation of a global pandemic [68,69,70].

Our results could be partially explained by the social role theory; namely, the lack of gender differences in mental health levels should imply a decrease in traditional male gender roles, equalizing the emotional expression patterns crucial for a person’s well-being. The above is supported by recent findings [71], showcasing the significant positive relationship between gender roles and psychological well-being, accompanied by self-acceptance. Additionally, men are inclined more toward learned helplessness, which could impair their mental health during a pandemic [72], leading to a lower result than they would have under regular circumstances.

Girls are prone to more critical thinking and rumination, and aligned with such findings are our observed lower levels of self-compassion, which have also been found in numerous previous reports [30,73]. In a Polish sample, significant differences in general self-compassion were demonstrated, with a higher score for men [74]. Furthermore, women had significantly higher scores in self-judgement, isolation, and overidentification, while men scored higher in mindfulness, which is similar to our results.

Significant gender differences in self-compassion have been found independently of mental health gender differences [75], or their absence, similar to our results. This could also signal that, while mental health contains at least partially biological foundations, self-compassion represents a cognitive scheme, mindset, or even emotional knowledge. Another interesting result is a recent survey [76] showing that masculine gender role stress was positively associated with self-stigma and self-coldness, but negatively associated with self-compassion, meaning that self-compassion serves as a protective factor when toxic masculinities, defined by some authors as authoritarian masculinity or an unrealistic image of masculinity, derived from estranged father–son relationships [77], discourage men from seeking help [78]. Yarnell found that gender role orientation and self-construal—or how (in)dependent individuals define themselves—moderate the relationship between gender and self-compassion [30]. They reported that women had slightly lower levels of self-compassion. However, further analysis revealed that those high in both femininity and masculinity tended to have the highest levels of self-compassion.

Our findings also indicate an overall positive attribution style in the entire sample. However, boys tended to score modestly higher (more negative) on three (out of four) subscales, including globality, stability, and negative consequences. Notably, the reliability score of the stability subscale, 0.534, makes it difficult to draw further conclusions. Such results might stem from the research design aiming to target general attributions rather than focusing on a more specific field. For instance, traditional social and cultural influences can also alter attribution biases, such as traditional gender roles regarding success and defeat in STEM fields [79], but there are optimistic results considering this, which indicate that gender discrepancies are decreasing [80]. Overall, gender differences in attributions can impact self-esteem, motivation, or even impostor syndrome [81], and our results indicate that young men could sometimes be an even more vulnerable group.

### 4.3. Correlations and Regression Model

The regression analysis model included predictor variables that significantly correlated with mental health as a criterion variable, four attribution subscales in the first step of the analysis, and six components of self-compassion in the second step. Stability and negative consequences explained 20% of the total mental health variance. Such results are aligned with comparable studies and indicate that adolescents with more stable negative attributions who expect more negative consequences might have poorer mental health. In the second step, we introduced variables related to self-compassion, which contributed to an additional 20.2% based on self-kindness, isolation, and common humanity. Both steps were significant and predicted 40.2% of the total variance in mental health. Significant predictors from the first step (stability and negative consequences) lost their impact once the variables related to self-compassion entered the analysis. The significance of common humanity and isolation as predictors of mental health suggests that individuals who see life’s difficulties as a part of a life experience that is not so rare have greater life satisfaction, feel accepted in their community, and are attached to their social groups, in return receiving more social support, both instrumental and emotional.

Self-compassion can moderate or even nullify the harmful effects of negative attributions, which is undoubtedly the most exciting finding thus far. Kawamoto et al. [82] found that self-compassion plays a significant role in dividing adaptive and maladaptive perfectionism; it has an adaptive function for perfectionists who inevitably need to accept their imperfections. According to Shin and Lim [7], self-kindness can affect individuals’ mental health, as registered in our correlation analysis. Self-kindness is, among other things, associated with a reduced risk of depression and anxiety, as well as increased psychological well-being, and reduces perceived stress, more so for individuals vulnerable to mental health problems [83]. Others [84] found the mediating effect of self-compassion on the relationship between religion/spirituality and well-being, as well as religion/spirituality and depression/anxiety symptoms. Research [34] conducted during the COVID-19 pandemic offers a different angle, where self-compassion did not moderate the relationship between the frequency of self-care and satisfaction with life or between the frequency of self-care and perceived stress levels, suggesting a need for a more complex model incorporating situational determinants.

Attributional theories are commonly used in models assessing several psychopathological phenomena and psychosocial health [85]. In our model, stability and perceived negative consequences explained one fifth of the total mental health variance, but their significance vanished once we added the self-compassion dimension into account. Negative consequences presume that other adverse events will follow, potentially causing worse mental health, while stability indicates how flexible individuals’ mental schema are [86]. It is possible that self-compassion incorporates both mental flexibility and optimism, thus reducing the effect of isolated cognition. The significant contribution of self-compassion and attribution in explaining mental health was confirmed in the present study, as well as a large number of previous studies [87,88,89]. Some reports during the pandemic even state how attributions shifted in dyadic (romantic) relationships, namely how maladaptive attributions increased in couples with lower functioning [89], implying the strong impact of the cognitive mental health dimension. Self-compassion was proven to enhance resilience, hope, optimism, productivity, and relationships, all of which are considered indicators of mental health [90]. Different samples and models also pointed out how the attribution process can generate an emotional response coinciding with mental health levels. For example, when exploring cyber aggression toward stigmatized groups during the pandemic, Wu and Zhang found indirect effects of self-compassion, mediated by cognitive pathways, i.e., attributions [91]. In a sample consisting of mental health nurses, positive attributes of self-compassion resulted in higher caring efficacy, which does not only indicate their mental health status, but could also predict the mental health outcomes of their patients [92]. In addition to everything mentioned above, rare longitudinal studies accentuate the importance of mental health by stating how baseline mental health levels are associated with all-cause mortality during an 18-year follow-up [93].

### 4.4. Limitations and Implications

The main limitation of this study is the data collection, which was conducted online, due to pandemic conditions. Despite being an economical and efficient method, it has some limitations, such as uncertainty about the participant’s real identity (including gender and other sociodemographic characteristics) and assessment conditions. The issues with Facebook-conducted surveys have been addresses in previous studies [94]. There was a large discrepancy in the sample size between male and female participants; thus, our findings must be taken with reservation. Next, the Cognitive Styles Questionnaire we used only consists of negative hypothetical events, meaning some participants could experience a priming effect for a more negative attribution style. Notably, it is saturated with cognitive skill items (particularly negative consequences and self-worth implications) and lacks one significant attribution dimension—internality. Internality would provide an answer for a causal dilemma; it is the source of the problem within an individual or an environment.

Offering tenderness and understanding to oneself even in the most challenging moments leads to self-acceptance and greater satisfaction with life. As Bluth and Neff [95] stated, self-compassion reduces psychopathology and increases well-being in both adolescents and adults. However, some authors have criticized the concept of self-compassion and its measures, stating that they are contaminated by psychopathological characteristics, vulnerability, and mental illness, and therefore multidimensional [96]. Based solely on our findings, self-compassion is portrayed as a valuable concept useful to designing interventions that could help older adolescents maintain or even improve their mental health, combining attribution theory settings and self-compassion workshops.

Many interventions have been tested and proven [29], but some [97] only found small to medium effects on reducing depression, anxiety, and stress in the immediate and follow-up tests. When inspecting the main differences in the supported studies, it becomes evident that we are in need of more experimental studies with control groups, and we would profit largely from inspecting the effect of gender roles. Additionally, not many participants in such studies had distress symptoms, meaning that we tend to focus on an already relatively healthy population, while lacking more clinical samples. Since the mental health of youths should be one of the societal priorities [98,99], we have to make a further effort to narrow down key factors that could serve as risk and/or protective factors and find new ways to introduce them into everyday mental health practices, especially considering the long-term consequences of the pandemic. As Wu at al. [100] pointed out, self-compassion was positively correlated with authentic, durable happiness, while mediation analyses indicated that the meaning of life partially mediated this association, which was proven as effective in the conditions of the COVID-19 pandemic. This could be one of possible valuable directions for future research and practice.

## 5. Conclusions

Based on the descriptive statistics, our sample covered functional, non-clinical older adolescents of both male and female genders. A gender difference analysis revealed no differences in positive mental health and less self-compassion in females, but also a tendency for boys to develop a more negative attributional style. Mental health was undoubtedly related to self-compassion and the attribution process. More specifically, participants with higher scores in the globality, negative consequences, self-worth implications, and stability categories had slightly impaired mental health, but the effect was not significant after the inclusion of self-compassion variables. In other words, older adolescents who are kinder to themselves, are less isolated, see failure as an integral part of the human experience, and accept a more growth-oriented mindset rather than a learned helplessness state will have better mental health. Although COVID-19-related variables did not prove to be significant, we must have in mind that we included only variables regarding personal experience and consequences of COVID-19, and that the entire social environment became very challenging during that time. Additional research is needed for future prevention and intervention programs targeting older adolescents.

## Figures and Tables

**Table 1 ijerph-20-06981-t001:** Descriptive data for all studied variables (N = 322).

Variable		Range	M	SD	S (SE)	K (SE)	K-S
Self-compassion	Self-kindness	1–5	2.91	0.89	0.109 (0.136)	−0.523 (0.271)	0.066 **
	Common humanity	1–5	3.00	0.83	−0.029	−0.527	0.084 **
	Mindfulness	1–5	3.13	0.89	−0.089	−0.579	0.073 **
	Self-judgment (rec.)	1–5	3.04	0.92	−0.064	−0.637	0.066 **
	Isolation (rec.)	1–5	2.94	0.94	0.037	−0.663	0.076 **
	Over-identification (rec.)	1–5	2.59	0.91	0.248	−0.636	0.088 **
	TOTAL	1–5	2.94	0.59	−0.120	0.445	0.038
Attributions	Globality	7–35	15.34	4.58	0.341	−0.434	0.082 **
	Stability	7–29	18.01	4.25	−0.341	0.271	0.081 **
	Self-worth implications	10–50	24.35	7.14	0.068	−0.125	0.045
	Negative consequences	3–15	6.46	2.48	0.553	−0.267	0.144 **
	TOTAL	27–135	64.17	15.81	0.181	−0.297	0.037
Well-being	Emotional well-being	0–5	3.02	1.20	−0.391	−0.531	0.111 **
	Social well-being	0–5	2.41	1.21	0.044	−0.875	0.074 **
	Psychological well-being	0–5	3.03	1.22	−0.381	−0.792	0.099 **
	TOTAL	0–5	2.81	1.11	−0.222	−0.841	0.083 **

Note: ** *p* < 0.01; S—skewness; K—kurtosis; SE—standard error (constant for all variables); KS—Kolmogorov–Smirnov goodness-of-fit test; rec.—recoded where higher score means higher self-compassion, regardless of the original direction of the subscale (e.g., higher score is shown for participants who expressed lower self-judgement).

**Table 2 ijerph-20-06981-t002:** Results of *t*-test for independent samples on all observed variables (df = 320).

		Boys (n = 76)	Girls (n = 246)	T	*p*
Self-compassion	M	3.08	2.89	−2.371	0.018
	SD	0.542	0.599		
Attributions	M	68.01	62.88	−2.444	0.015
	SD	14.625	14.699		
Mental health	M	2.96	2.76	−1.248 * (df = 109.177)	0.215
	SD	1.269	1.058		

Note: * Equal variances not assumed in *t*-test.

**Table 3 ijerph-20-06981-t003:** Frequency of participants’ experience with COVID-19-related issues during the pandemic.

Item	Answer	n	*p*
Have you or has someone close to you had the coronavirus?	YES	212	65.8%
	NO	110	34.2%
If so, did you have any health consequences as a result?	YES	21	15.7%
	NO	113	84.3%
Are you or is someone close to you in a high-risk group for contracting the coronavirus?	YES	236	73.3%
	NO	86	26.7%
Have you or has someone close to you been hospitalized due to the coronavirus?	YES	51	15.8%
	NO	271	84.2%
Has someone close to you died from the coronavirus?	YES	30	9.3%
	NO	292	90.7%
Have you or has someone close to you lost their job due to the coronavirus?	YES	55	17.1%
	NO	267	82.9%

**Table 4 ijerph-20-06981-t004:** Results of Mann–Whitney U test for independent samples based on COVID-19 experiences during the pandemic (N = 322).

Variable		Mean Rank “YES”	Mean Rank “NO”	MWU	*p*
Infected by coronavirus	Self-compassion	157.12	169.95	12.589	0.241
	Mental health	160.31	163.80	11.912	0.750
	Attributions	162.88	158.83	11.366	0.711
Health consequences	Self-compassion	69.14	67.19	1.152	0.833
	Mental health	71.36	66.78	1.105	0.620
	Attributions	76.29	65.87	1.002	0.259
High-risk group	Self-compassion	161.46	161.42	10.158	0.989
	Mental health	164.26	153.91	9.495	0.377
	Attributions	160.19	165.09	10.457	0.676
Hospitalization due to COVID-19	Self-compassion	159.29	161.92	7.023	0.854
	Mental health	165.09	160.82	6.705	0.737
	Attributions	165.52	160.74	6.727	0.764
Death due to COVID-19	Self-compassion	165.30	161.11	4.409	0.952
	Mental health	159.48	161.71	4.440	0.901
	Attributions	165.30	161.11	4.266	0.814
Job lost due to COVID-19	Self-compassion	182.30	157.22	8.604	0.045 *
	Mental health	157.90	162.24	7.540	0.753
	Attributions	182.30	157.22	6.198	0.069

Note: * *p* < 0.05; MWU = Mann–Whitney U nonparametric tests for independent samples.

**Table 5 ijerph-20-06981-t005:** Pearson’s correlation matrix for predictor and criterion variables (N = 322).

	1	2	3
Self-compassion (1)	1	−0.456 **	0.560 **
Attributions (2)		1	−0.436 **
Mental health (3)			1

Note: ** *p* < 0.01.

**Table 6 ijerph-20-06981-t006:** Correlation matrix for multiple regression analysis (N = 322).

	1	2	3	4	5	6	7	8	9	10	11
Self-kindness (1)	1	0.577 **	0.723 **	0.404 **	0.089	0.052	−0.385 **	−0.272 **	−0.332 **	−0.320 **	0.488 **
Common humanity (2)		1	0.635 **	0.144 **	0.016	−0.006	−0.272 **	−0.203 **	−0.228 **	−0.213 **	0.422 **
Mindfulness (3)			1	0.135 *	−0.002	0.006	−0.330 **	−0.162 **	−0.231 **	−0.284 **	0.453 **
Self-judgment (4)				1	0.659 **	0.664 **	−0.289 **	−0.343 **	−0.338 **	−0.290 **	0.316 **
Isolation (5)					1	0.681 **	−0.195 **	−0.280 **	−0.292 **	−0.249 **	0.311 **
Over-identification (6)						1	−0.081	−0.218 **	−0.161 **	−0.152 **	0.232 **
Globality (7)							1	0.532 **	0.721 **	0.728 **	−0.391 **
Stability (8)								1	0.609 **	0.460 **	−0.351 **
Self-worth implications (9)									1	0.633 **	−0..370 **
Negative consequences (10)										1	−0.388 **
Mental health total (11)											1

Note: * *p* < 0.05; ** *p* < 0.01.

**Table 7 ijerph-20-06981-t007:** Multiple regression analysis of components of self-compassion and elements of attributions as predictors of mental health.

	Step One	Step Two
	Β	Β
Globality	−0.149	−0.065
Stability	−0.165 **	−0.090
Self-worth implications	−0.092	−0.034
Negative consequences	−0.158 **	−0.102
Self-kindness		0.231 **
Common humanity		0.162 **
Mindfulness		0.126
Self-judgment		−0.107
Isolation		0.225 **
Over-identification		0.093
R	0.448	0.634
R2	0.200	0.402
∆R	-	0.202
F	19.864 **	20.903 **

Note: ** *p* < 0.01; R = multiple correlation coefficient; R2 = coefficient of multiple determination; ∆R = change size for R2; F = F-test of overall significance of regression model.

**Table 8 ijerph-20-06981-t008:** Multiple regression analysis coefficients.

		B	SE	β	T	*p*	Tolerance	VIF
Step 1	(Constant)	4.813	0.255		18.886	0.000		
	Globality	−0.031	0.021	−0.127	−1.504	0.134	0.352	2.844
	Stability	−0.042	0.017	−0.160	−2.489	0.013	0.610	1.640
	Self-worth	−0.011	0.012	−0.069	−0.859	0.391	0.394	2.539
	Negative consequences	−0.080	0.034	−0.178	−2.359	0.019	0.445	2.248
Step 2	(Constant)	1.196	0.446		2.681	0.008		
	Globality	−0.012	0.018	−0.051	−0.671	0.503	0.333	2.999
	Stability	−0.027	0.015	−0.103	−1.797	0.073	0.584	1.713
	Self-worth	−0.002	0.011	−0.015	−0.210	0.834	0.384	2.606
	Negative consequences	−0.050	0.030	−0.111	−1.666	0.097	0.436	2.296
	Self-kindness	0.309	0.098	0.247	3.159	0.002	0.314	3.188
	Common humanity	0.205	0.079	0.152	2.601	0.010	0.561	1.784
	Mindfulness	0.160	0.091	0.128	1.755	0.080	0.364	2.749
	Self-judgement	−0.160	0.094	−0.133	−1.712	0.088	0.318	3.143
	Isolation	0.277	0.079	0.234	3.512	0.001	0.433	2.311
	Over-identification	0.126	0.084	0.103	1.497	0.136	0.408	2.449
a. Dependent variable: mental health								

Note: B = unstandardized Beta weights or regression coefficients; SE = the standard error of the regression; β = standardized Beta; T = *t*-test statistics; *p* = probability value; Tolerance = indicator of multicollinearity; VIF = variance inflation factor.

## Data Availability

The data presented in this study are available on request from the corresponding author. The data are not publicly available due to lack of funding.

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
