# Peer review of "The Role of Self-Compassion and Attributions in the Mental Health of Older Adolescents amid the COVID-19 Pandemic"

_ijerph, 2023, doi:10.3390/ijerph20216981_

Round 1

Reviewer 1 Report

Comments and Suggestions for Authors

The authors studied self-compassion, attribution style, and general mental health in young adults.  The paper was generally well-written, but would require significant editing to better define and execute the authors' goals.  I appreciate their efforts and hope they find my feedback helpful.

-I would recommend that the authors seek assistance with the academic tone of their writing.  This may also involve help with english language.  

-The paper would benefit from editing (e.g., attention to proper punctuation).

-The authors aims are unclear.  I was unsure if they are attempting to describe mental health in this population in general, or if they are aiming to investigate mental health as it relates to the pandemic.  It will be important to tighten up their aims.

-Related, while very thorough, the review of literature in the introduction section is very broad and general.  It will be important for the authors to target their review why it is important to study their selected variables during the time frame they chose (pandemic).  It is important to highlight what is known and what is missing in current literature to provide more justification on why their study is needed and what it adds to our research base.  Much more review is needed on how their variables relate to the pandemic.

-In the introduction, the authors reviewed these variables as they relate to adolescents, yet their participants were young adults (ages 18 - 22).  I would recommend adjusting their lit review to target young adults rather than adolescents.

-It will be important to clarify their aims.  Specifically, clarify whether they are looking to explain components of mental health in general or components of mental health during the pandemic.  Currently there are too many unclear aims.  It will also be helpful to better explain what they mean by "contribution of elements" in regards to mental health (pg. 3, line132).

-Please include the inclusion/exclusion criteria for their sample.  It would also be helpful for the reader to have more detail about the recruitment procedures.  For better flow, it would be nice to describe this process before describing participant characteristics.

-Was the sociodemographic scale developed by the authors? If so, it would be helpful to describe this process.  It would also be helpful to provide copies of this scale or survey in supplemental materials.

-Please provide cut off points or indications for what ranges of scores mean for each questionnaire (e.g., higher scores mean greater self-compassion).  Some of this description was in the results section (pg 6., beginning line 226), but the results section should really be focused on the data.  

-The results section would benefit from editing for academic presentation.

-It would be helpful to have p values included in Table 2

-It is difficult to understand results on Covid status.  I am unsure what COVID-related variables were studied.  It would be helpful to have a table for these results, in addition to a more detailed description of the variables in the Methods section.

-It is helpful to star significance (at .01 and .05 levels) on all tables.

-The discussion section did not cover pandemic-related findings.  This should be included as a stated aim. 

-Much of the discussion section included review of other literature.  It is very helpful to first provide a general overview of results, and then focus this section on the implications of their results.  What is now known about Croatian youth and these components of mental health that was not previously known?  What should future research continue to investigate?

Comments on the Quality of English Language

The paper was generally well written, but would benefit from attention to organization/flow, academic tone (e.g., presenting results, describing statistical processes, phrasing), and light editing (some punctuation missing, some spelling errors).

Author Response

Dear Reviewer,

We thank you for your time and attention in reviewing the article as well as your remarks and suggestions provided so that we could improve the quality of the paper and its publication potential. Several changes have been made to the content in order to clarify the objectives of the study and the main results obtained. We have since added an additional statistical analysis to further enlighten the COVID-19-related issues. Also, we added 57 new references and deleted 45 from the original version, and as suggested by both reviewers, opted to use MDPI's Language Editing Services. They also sugested minor change of the title, that we excepted. The discussion section has been reviewed and we incorporated some of the suggestions/questions that you present, which gave us the opportunity to reflect.  Since we made such a major revision, we haven't marked all the changes in text, but we marked all added references in the reference list. We enlisted all the changes in a separate document, indicating line numbers where the changes have been made. In the hope this effort gets recognized and makes us eligible for your publication, 

we offer you our kind regards,

                                                                            Authors

Reviewer 2 Report

Comments and Suggestions for Authors

This study examines the relationship among mental health, self-compassion, and attributions  during COVID-19 in an online study of 333 Croatian teenagers between 18-22 years of age to determine the role of self-compassion in predicting mental health. For the most part, the results are supportive of previous studies regarding these three variables with respect to the positive role of self-compassion noted. What differs is the there was no notable difference between girls and boys concerning self-compassion as had been found in the past.

The strengths of this paper are the extensive statistical analyses that were conducted to reach conclusions and the logical progression of the process reported. At times, the writing style is very odd for a native English speaker. However, the greatest concern for a study of COVID-19 conditions is that few of the references were published during COVID-19. The authors have not recognized the importance of COVID-19 specific research to support their claims. An additional problem is that, although part of what the authors were intending to investigate was the role of gender with respect to self-compassion, because of the online nature of the study, they actually could not be sure of the gender of their participants. As a result, the research results are truly called into question. Below are the line by line comments and suggested edits.

Line by line suggested edits.

7 Change “between” to “among”.

29 There is no actual introduction provided to this Introduction. Instead, the authors immediate present three subsections (mental health, self-compassion, and attributions) without introducing the point of the paper and that this reason divides the  Introduction into three subsections in relation to that focus. Therefore, please add an opening paragraph to this Introduction indicating the purpose of the paper, that there will be three subsections in relation to the purpose in the Introduction, and why it is important to have these subsections given the purpose of the paper.

30 Given that the definition cited of Keyes is from 2005, the authors must explain why, in a science paper—where references should be within the past five years—it should matter that the definition by Keyes be considered still relevant.

32 Why have the authors qualified the statement with “Even”? Change “Even” to “As well,”

34 Again, the citation to Galderisi et al. is to a paper published in 2015. Why have the authors chosen a reference that is so old? If there is a reason, the authors must explain it in the text. If there is no real reason, the authors are asked to find a current reference.

30-37 This entire first paragraph is very weak. The first sentence focuses on flourishing. The next is in regards to being productive, while the third contradicts the second saying that being productive is not a good indication of mental health because of contextual factors. In the end, what is the point of the first paragraph? Is it to say that only flourishing is a measure of mental health because being productive is not always an indication? The authors must be clear on the point they are trying to make with the first paragraph.

38-43 Similarly, with the second paragraph, the first two sentences regarding the relationship between mental health and mental illness have nothing to do with the final sentence that presents three dimension of positive mental health. What is the purpose of the first two sentences in relation to the last? If there is a relationship, the authors need to explain in the text what it is. As well, citations 5 ad 6 are to older works by Keyes. Here is a Google Scholar search of CLM Keyes for publications since 2019; notice that there are many to choose from. The authors are asked to please support their claims with recent research by Keyes: https://scholar.google.ca/scholar?as_ylo=2019&q=CLM+Keyes&hl=en&as_sdt=0,5

44 Although in 2023 readers may know what “the pandemic” means, the authors need to explain in greater detail to what pandemic they are referring and include references regarding the pandemic.

45-49 Many of the descriptive words chosen in these sentences are unusual for a native English speaker to the extent that they are inappropriate for a science paper. Please consult science writer who is a native English speaker for better word choices in these sentences.

61-92 Of the twenty citations in this subsection, ten are out of date. Please substitute the out of date references for ones that are current.

61-66 The authors begin the first paragraph of this subsection referring to self-esteem. However, they don’t say what is the relationship between self-esteem and self-compassion. Yet, in line 66, they begin to discuss self-compassion as if it were self-evident how self-esteem and self-compassion relate. The authors need to indicate what is the relationship between self-esteem and self-compassion. As well, they need to specify why it is important to discuss self-esteem when referring to self-compassion.

94-125 Of the twelve references in this subsection, only three are current. This is not acceptable for a science paper. Please find current references to support the claims made in this subsection.

110 It is unclear how the authors have moved from a discussion of adverse events to a theory of helplessness that has been upgraded to hopelessness. The authors must include connecting sentences so that the reader can see the relationship.

118-120 What does self-compassionate people accepting responsibility have to do with positive mental health? The authors need to provide connecting sentences to demonstrate the relationship.

123 What is “control belief about learning”? This does not make sense in English.

127-138 There are three citations in this paragraph—two of them are very outdated. This is especially problematic in a paragraph concerning the aim of this research in relation to COVID-19 status. Please update the supporting references to current research.

127-130 This sentence belongs with the previous subsection on self-compassion.

130 Change “Meanwhile, the” to “The”.

141 When and where were the participants recruited? Please state this in the text.

154 How was informed consent obtained from the participants? Please state this in the text.

157-159 Each of the questionnaires used were ones developed pre-COVID-19, yet, the authors have not demonstrated that the questionnaires they selected were appropriate for use during COVID-19. Please cite references to studies done during COVID-19 that made use of these same questionnaires.

174-178 Each of the citations are to very old references. The authors need to provide COVID-19-related references to demonstrate the continuing value of these questionnaires.

189 The authors have focused on the early work of CLM Keyes. Here is a COVID-19-related reference including CLM Keyes as an authors. The authors need to determine if this paper supports the 2002 work of Keyes for use in COVID-19 studies:

Fuller-Thomson, E.; Lung, Y.; West, K.J.; Keyes, C.L.M; Baiden, P. Suboptimal baseline mental health associated with 4-month premature all-cause mortality: Findings from 18 years of follow-up of the Canadian National Population Health Survey. J. Psychosom. Res2020136, 110176. https://doi.org/10.1016/j.jpsychores.2020.110176

201 The authors likely mean “approval” rather than “Statement”. What is important to know is that the authors obtained approval for their research. Please state that approval was obtained.

202 Please explain in the text why the authors opted to conduct their research via social networks and why they made the particular choice in social networks they did.

206 Was there any incentive to participate in the research? Please answer this in the text.

208-210 Please explain why each of these methods of analysis were chosen and provide COVID-19-related references to support their use for research conducted during COVID-19.

219-220 Why are there two sets of data for “Self-kindness”? Please explain in the text. Please also explain in the text what “(rec.)” means. 

232 By “flourish” and “wither in life”, what do the authors mean? Please use terms that are descriptive and appropriate for a science paper.

234-251 The information in the text does not discuss the results of Table 2. Instead, it introduces other results that are not recorded in Table 2. Either record the results referred to in the text in Table 2 or stick with describing the results found in Table 2.

258-259 Where is Mean Rank indicted in the tables? Please explain in the text.

287-289 Table 5—Please specify in the legend the meaning of R, R2, ∆R, and F.

291-292 Table 6—Please indicate the meaning of each of the column headings in the legend.

317 The authors have cited a paper from 2009 as if the reference concerned today. Please find a current reference to support the claim.

325 Citation 25 is from 2015. It cannot be used to imply what is known currently. Please update the reference.

329-360 Of the ten references in these lines, only three are current. Please find current references to support each of the claims.

340 Need a current references regarding gender fluidity.

345 The authors claim that “not a lot of recent studies can be found on gender differences in cognitive styles regarding attribution process”. Here is a Google Scholar search of research that has been done of this topic since 2019 that returned about 16,900 results. Please rewrite this claim and references some of these papers. https://scholar.google.ca/scholar?as_ylo=2019&q=gender+differences+in+cognitive+styles+regarding+attribution+process&hl=en&as_sdt=0,5

362-387 In a subsection regarding COVID-19, not one of the citations is to studies done during COVID-19. This is unacceptable. KD Neff has published related to COVID-19 (https://doi.org/10.1080/17439760.2021.1871945). Please cite this paper.

393 “comparable studies”—please cite current comparable studies.

399-403 Please provide at least one citation to a current reference to support these claims.

405-428 Of the nine references in these paragraphs, only three are current. Regarding COVID-19-related research, all references must be to research conducted during COVID-19. Please updated the references.

442 Citation 75 is to an out of date reference. Please find a current reference to support this claim.

447 Citation 47 is to a 2014 publication. It can have nothing to say about COVID-19 conditions. Please find a COVID-19 reference to support this claim.

483-Please redo all the references according to MDPI style—see the Word template or the Instructions for Authors.

Comments on the Quality of English Language

There are a number of concerns regarding the English, both with respect to words chosen and the structuring of sentences. These issues are pointed out to the authors in the Comments and Suggestions for Authors.

Author Response

(The authors gave the same response as above.)

Round 2

Reviewer 2 Report

Comments and Suggestions for Authors

The authors are thanked for the changes they have made. Particularly the deletion of old references and addition of references published since the advent of COVID-19 are important to the value of the submission. The new paragraphs that the authors have added have greatly improved the interest level of the piece. They are well-researched and well-constructed.

Please check the Instructions for Authors on the IJERPH website and redo the references in MDPI style.

Line by line suggested edits

53 Change “one” to “people”.

110-112 Need a current reference to support this claim.

117-119 Need a current reference to support this claim.

131 Change “help in” to “help when in”.

132 Change “proactivity” to “engaging in physical activity”.

215 Change “Subsequently” to “As such”. Also, add this paragraph to the one before.

228 Change “pupils” to “students” (pupils implies they were in elementary school).

230-232 Please provide the information on how many cities and towns were included in the study.

261 Change “an individual towards himself” to “individuals towards themselves”.

262-263 Please provide additional information on how the authors’ developed this theoretical framework including references to previously published research on this framework. If none has been published, the authors need to provide the method used in developing this framework.

273 The authors have not demonstrated that this very short form has been used in COVID-19-relataed research. Here is a reference they can use in this regard.

Schuster, R.; Fischer, E.; Jansen, C.; Napravnik, N.; Rockinger, S.; Steger, N.; Laireiter, A. R. Blending Internet-based and tele group treatment: Acceptability, effects, and mechanisms of change of cognitive behavioral treatment for depression. Internet Intervent 202229, 100551. https://doi.org/10.1016/j.invent.2022.100551

290 The authors have not demonstrated the short form has been used in COVID-19-related research. Here is a Google Scholar search with a number of returns regarding the use of this scale during COVID-19. Pick the most appropriate and reference it: https://scholar.google.ca/scholar?hl=en&as_sdt=0%2C5&q=mental+health+continuum+short+form+COVID-19&btnG=

307-308 Please provide a supporting reference for claiming that Facebook was the most used social network in Croatia at the time.

318 Although the authors have stated in their response to the reviewer that other articles omit why they are using their statistical software this does not mean it is a good practice to follow. Here is a Google Scholar search that provides a number of returns to indicate that this statistical software has been used regularly with respect to COVID-19 data in regards to similar studies. Please pick the most appropriate ones and reference them: https://scholar.google.ca/scholar?hl=en&as_sdt=0%2C5&q=IBM®+SPSS®+Statistics+software%2C+version+29+COVID-19+research&btnG=

383-384 The column between “Item” and “N” requires a heading.

387 Please reference the Mann–Whitney U nonparametric test and show that it has been used in previous COVID-19-related research. Here is a Google Scholar search with a number of returns in this regard. Please pick one and reference it: https://scholar.google.ca/scholar?hl=en&as_sdt=0%2C5&as_ylo=2019&q=Mann–Whitney+U+nonparametric+test+COVID-19&btnG=

400 Please explain in the text why you thought to do this inspection.

480 Although the copyeditor may have suggested the word “sympathy”, this word is now only used in relation to the death of an individual. “Empathy” is the word that is used in its place. Please change “sympathy” to “empathy”.

482 Change “stricter to themselves” to “stricter with themselves”—this is the usual way of phrasing it. Change “must admit” to “must accept” (there is a need for acceptance, not for admission).

484-486 Change “experiences, for example, in cases of unemployment and reemployment; the perception that “chance” was the controlling factor predicted longer unemployment” to “experiences. For example, the perception that “chance” was the controlling factor predicted longer unemployment in cases of unemployment and reemployment”.

495-498 Change “research. As we stated earlier, findings concerning that issue are diverse, although it is predominantly shown that young women are more vulnerable, especially  in the situation of a global pandemic” to “research, in contrast to the predominant finding of previous research that young women are more vulnerable in the situation of a global pandemic”.

510 Change “proven” to “demonstrated”.

514 Change “were found” to “have been found”.

515 Change “like in our results” to “similar to our results”.

516 Change “signalize” to “signal”.

518 Change “cue” to “result”.

521 “toxic masculinities” needs to be defined and referenced.

526-568 Please provide a current peer-reviewed reference for this further analysis.

538 Change “in entire sample” to “in the entire sample”.

640 Please cite some of these “large number of previous studies”.

656 Change “Implications and limitations” to “Limitations and implications”.

675-668 Please also note in this regard that because the study was conducted online the authors cannot be sure that the genders provided by the participants were their actual genders. Especially in Facebook, young people are known to fake information in their profiles, including their gender. Here is a reference:

Albayati, M. B.; Altamimi, A. M. An empirical study for detecting fake Facebook profiles using supervised mining techniques. Informatica 201943. https://doi.org/10.31449/inf.v43i1.2319

686 “the mental health of youths should be one of the societal priorities”—please provide current peer reviewed references supporting this imperative.

Comments on the Quality of English Language

The quality of the English is much better, especially in the entirely new paragraphs. However, there are still a number of places where the English is incorrect. These have been noted in the Comments and Suggestions for Authors.

Author Response

Thank you very much for taking the time to review this manuscript. Please find the detailed responses in the attachment and the corresponding revisions in track changes in the re-submitted files.
